# Optimising Seasonal Streamflow Forecast Lead Time for Operational Decision Making in Australia

Andrew Schepen[1], Tongtiegang Zhao[2], Q.J Wang[2], Senlin Zhou[3] and Paul Feikema[3]

[1]CSIRO Land and Water, Dutton Park, 4102, Australia
[2]CSIRO Land and Water, Clayton, 3168, Australia
[3]Bureau of Meteorology, Melbourne, 3001, Australia

*Correspondence to*: Andrew Schepen (Andrew.Schepen@csiro.au)

**Abstract.** Statistical seasonal forecasts of three-month streamflow totals are released in Australia by the Bureau of Meteorology and updated on a monthly basis. The forecasts are often released in the second week of the forecast period, due to the onerous forecast production process. The current service relies on models built using data for complete calendar months, meaning the forecast production process cannot begin until the first day of the forecast period. Somehow, the Bureau needs to transition to a service that provides forecasts before the beginning of the forecast period; timelier forecast release will become critical as sub-seasonal (monthly) forecasts are developed. Increasing the forecast lead time to one month ahead is not considered a viable option for Australian catchments that typically lack any predictability associated with snowmelt. The Bureau's forecasts are built around Bayesian joint probability models that have antecedent streamflow, rainfall and climate indices as predictors. In this study, we adapt the modelling approach so that forecasts have any number of days lead time. Daily streamflow and sea surface temperatures are used to develop predictors based on 28-days sliding windows. Forecasts are produced for 23 forecast locations with 0–14 and 21-days lead time. The forecasts are assessed in terms of CRPS skill score and reliability metrics. CRPS skill scores, on average, reduce monotonically with increase in days of lead time, although both positive and negative differences are observed. Considering only skilful forecast locations, CRPS skill scores at 7-days lead time are reduced on average by 4 percentage points with differences largely contained within +5 to -15 percentage points. A flexible forecasting system that allows for any number of days of lead time could benefit Australian seasonal streamflow forecast users by allowing more time for forecasts to be disseminated, comprehended and made use of prior to the commencement of a forecast season. The system would allow for forecasts to be updated if necessary.

## 1 Introduction

The Australian Bureau of Meteorology (the Bureau) operates a statistical seasonal streamflow forecasting service to assist water management agencies make informed decisions about water management strategies in the season ahead. The forecasts provide probability distributions of the total volume of streamflow over the next three-month period. The forecasts are used by a variety of groups, including federal, state and local governments and their agencies, such as water management authorities;

agriculture and water management sectors; private businesses, general public and local communities. The forecasts help reduce the uncertainty in seasonal flow volumes to be expected, and therefore provide users with more certainty in decision making. For example, water authorities use forecasts to assist decisions on water allocation and water restrictions, manage river operations, schedule environmental watering, develop water transfer strategy and provide water order advice. They also use

forecasts to guide future storage levels, help decide on releases, produce water allocation outlooks to inform water markets, and manage risks at construction sites along rivers. State government agencies use forecasts to make decisions about environmental monitoring of streams, schedule irrigation, assess flood potential, and determine agency resourcing and public messaging.

The forecasts are currently updated once per month. However, the forecasts are frequently not released until the second week

of the forecast season because the forecast production process is overly time-consuming. Feedback from a user survey showed many users would prefer the forecasts to be issued earlier in the month, which would fit in better with their reporting schedules and operational timelines. The need to have forecasts issued earlier becomes more pronounced when issuing a one-month forecast rather than a three-month forecast.

The first step in the forecast process is to gather predictor data and perform quality control. Once the predictor data are assured

to be of good quality, forecasts and forecast products are generated. The forecasts subsequently need to be checked for modelling errors and inconsistences. Once all the forecast products are ready, key messages and other communication products are developed. Once the communication strategy is in place, the forecasts are released. The Bureau now issues forecasts for over 200 locations and so, understandably, the forecast production process is a time-consuming one.

Currently, the statistical forecasting models rely on predictor data observed up to the day prior to the first day of the forecast

period. For example, a forecast for the austral spring (September–October–November) requires observed data up to (and including) the 31st of August. The forecasts effectively have no lead time and so the forecast lead time can be considered to be 0-days. Suppose predictor data were available immediately on the first day of the forecast period; then forecasts could be generated on the first day of the forecast period. Even in this case, the forecast release would occur several days into the forecast season, after forecast products and communication messages were created. In reality, by the time the forecasts are

released, the forecasts typically have a lag time of at least 7 days.

The lag time in forecast release diminishes the value of the forecasts. Ideally, the forecasts would be in the hands of decision makers well ahead of the forecast season. Furthermore, the lag time in forecast release is likely to become more prominent in the short term because the Bureau plans to release sub-seasonal (i.e. monthly) streamflow forecasts by 2017. It is therefore important to investigate ways to issue the forecasts earlier and to quantitatively analyse how forecast accuracy and reliability

are affected.

The Bureau's current forecasting system is built around the Bayesian joint probability (BJP) modelling approach (Wang and Robertson, 2011;Wang et al., 2009). Although other statistical methods have been previously investigated for seasonal streamflow forecasting in Australia (e.g. Westra et al., 2008;Piechota et al., 2001, 1998;Chiew and Siriwardena, 2005), the BJP approach was initially adopted for operations owing to its wide potential applicability to perennial and ephemeral

catchments, in cases of pervasive missing data and/or pervasive zero flows and in situations where multiple sources of predictability were identified. BJP models make use of two types of predictors: predictors representing initial catchment conditions and predictors representing the climate state. The predictors are selected following a rigorous predictor selection process (Robertson and Wang, 2012). Initial catchment condition predictors, which act as a soil-moisture proxy, include

streamflow totals over the preceding 1, 2 or 3 months. Predictors representing the climate conditions are monthly climate indices lagged up to 3 months (Schepen et al., 2012;Kirono et al., 2010) and include indices representing the El Niño Southern Oscillation, variability in the Indian Ocean and variability in the southern polar circulation.

Predictors in the BJP modelling approach vary by season and by catchment. Therefore in operational forecast production, it is necessary to prepare a wide range of predictor data from a variety of data sources. Since the Water Act (2007) was introduced

in Australia, the Bureau of Meteorology is authorised to collect streamflow data from gauge owners. Streamflow data is therefore easily obtained for most gauges; however, the data comes in a multitude of formats and needs to be processed by data managers prior to being made available to forecasters. Usually, daily streamflow data for each of the forecast locations is available with a few days lag. Climate data is sourced from the Bureau of Meteorology and the United States National Oceanic and Atmospheric Administration (NOAA). NOAA ERSST (Huang et al., 2015) monthly sea surface temperature (SST) grids

are used for calculation of climate indices such as Nino3.4 and the IOD mode index. The ERSST data set for the previous month is normally available by the 4th day of the month. Other indices, such as the Antarctic Oscillation Index (AAO) (Mo, 2000) representing the Southern Annular Mode can take up to a week to become available. The Bureau of Meteorology updates Southern Oscillation Index values relatively swiftly, normally within a couple of days.

As just described, up to 7 days delay in the forecast production process can be attributed to delay in acquiring predictor data.

Much of the problem lies in the reliance on data for complete calendar months. An immediate and obvious resolution to the problem is to produce forecasts with 1-month lead time. Forecasts with 1-month lead time are certainly feasible in a statistical framework and are operationally useful in many parts of the world where snow-melt is a major source of seasonal streamflow predictability, e.g. the western United States (e.g. Pagano et al., 2004) and the Three Gorges system in China (Xu et al., 2007). However, increasing the forecast lead-time for Australian catchments beyond one month is undesirable because the primary

source of skill is initial catchment conditions and catchments have limited memory or short response times.

An alternative approach to get forecasts to users in good time is to make use of daily streamflow and climate data to generate predictors and forecasts with any number of days lead time. Although it could be presumed that skill will tend to be reduced as lead time increases, it is not known to what degree skill will be impacted. The optimal forecast lead time (in days) corresponds not only to the most skilful forecast that can be released prior to the beginning of the forecast period, but needs to

allow for forecast reliability, forecast preparation and communication time, and giving enough time for users to ingest streamflow forecasts into their models ahead of the forecast season.

In this study, we develop BJP models to produce forecasts with up to 21 days lead time for 23 catchments across Australia and seek to demonstrate that it is possible to release operational forecasts ahead of the commencement of the forecast target season. We investigate the availability of daily climate data and establish necessary modifications to the predictor choices and length

of record used to establish the models. Cross-validation forecasting experiments are conducted to evaluate the quality of forecasts. The skilfulness of forecasts are compared for 0–14 and 21 days lead time. The results give an estimate of how forecast skill changes with increasing forecast lead time. Subsequently, the optimal lead time is determined, based on considerations of forecast skill, data availability and forecast preparation time.

The remainder of the paper is organised as follows. Section 2 describes the study forecast catchments, streamflow and climate data. Section 3 details the study methods, including information about the BJP modelling approach and forecast verification. Section 4 presents the results. Section 5 discusses the results. Section 6 concludes the paper.

## 2 Catchments and Data

### 2.1 Catchments

Twenty-three forecast locations are selected for this study including forecast locations in all states: Queensland, New South Wales, Victoria, Tasmania, South Australia, Western Australia; plus the Northern Territory. Flows at the forecast locations are a mixture of perennial, ephemeral and intermittent flows. Table 1 summarises information about the forecast locations, including long catchment name, short ID, state (in the jurisdictional sense), upstream catchment area and approximate centroid latitude and longitude. The catchments range in size from 102–36,230 km$^2$. Furthermore, the forecast locations are plotted on

a map of Australia in Figure 1, which also indicates whether the catchment is in a temperate, subtropical or tropical climate zone.

### 2.2 Streamflow data

Daily streamflow data for the period 1982–2012 is sourced from the Bureau of Meteorology for all 23 forecast locations. The Bureau sources the data from the respective data owners and performs quality control. Daily streamflow data records often

contain missing values, for example, due to the failure of gauging equipment. The Bureau partially infills missing data records. Records for all forecast locations are infilled by linear interpolation if the number of missing days is equal to or less than 3 days. For some forecast locations, records are infilled by linear regression with nearby forecast locations if the number of missing days is equal to or less than 14 days.

In this study, the daily streamflow data are aggregated to 28 day totals for use as predictors and three-month totals (always

beginning on the first day of the month) for use as predictands. Many of the data records still contain missing data after the infilling process, however the proportion of missing data is never greater than 10%.

### 2.3 Climate Data

The Bureau's current BJP seasonal streamflow forecasting models rely on predictors including surface and subsurface ocean temperatures, the Southern Oscillation Index (SOI) and the Antarctic Oscillation Index (AAO). The predictors are identified

through a rigorous predictor selection process (Robertson and Wang, 2012). In this study, we adopt the climate predictors used

by the Bureau in their operational models with some changes. The biggest change is that we restrict the set of climate predictors to sea surface temperature (SST) predictors, substituting an ENSO SST index wherever the climate predictor in the Bureau's operational model is subsurface ocean temperatures or SOI. SST climate predictors representing ENSO and the Indian Ocean state are likely to be stable across a period of several weeks and are known to have strong lagged relationships with Australian

rainfall up to three months ahead (Schepen et al., 2012). Therefore, the predictors are likely to remain valid predictors as forecast lead time is increased by a few weeks. On the other hand, it is less certain that the AAO, representing the Southern Annular Mode, will remain a valid climate predictor as lead time is increased since it has a weak lagging relationship with Australian seasonal rainfall (Schepen et al., 2012). In cases where AAO is a climate predictor in the Bureau's operational model, no climate predictor is used in this study.

Daily SST data is obtained from the NOAA 1/4° daily Optimum Interpolation Sea Surface Temperature (daily OISST) (Reynolds et al., 2007). Full years of data are available from 1982 onwards. The daily OISST is constructed by combining data from satellites, ships, buoys, probes and ocean-dwelling robots. Interpolation fills in missing gaps and ensures a complete historical record with no missing data. Compared to monthly ERSST records, which go back to 1854, the daily OISST record is relatively short but it is necessary for generating timelier forecasts. To obtain SST climate predictors we calculate 28-day

average sea surface temperatures relative to 1982-2010 climatology. Area averaging is used to obtain the set of climate predictors: Nino3.4, Nino3, Nino4, the El Nino Modoki Index (Ashok et al., 2007), the Indian Ocean Dipole (Saji et al., 1999) and the Indonesian Index (Verdon and Franks, 2005). Figure 2 demonstrates that the monthly Nino3.4 values calculated from daily OISST and NOAA's monthly ERSSTv4 product are almost identical for the period 1982-2011 with $R^2 = 0.98$.

## 3 Methods

### 3.1 BJP forecasting models

Forecasting models are set up using the Bayesian joint probability (BJP) modelling approach (Wang and Robertson 2011; Wang et al. 2009). Separate forecasting models are established for each forecast location (23 locations), season (12 three-month seasons from the start of each month) and for 0–14 and 21 days lead time (16 lead times).

The BJP forecasting models make use of two types of predictors: predictors representing initial catchment conditions and

predictors representing the climate state. For all models, the initial catchment condition predictor is fixed to the previous 28 days total flow volume. This predictor will not be optimum for all forecast locations; however, it is a pragmatic choice that is likely to represent the overall catchment wetness reasonably well. Additionally, it gives a consistent model set up without needing to undertake initial catchment condition predictor selection. SST climate predictors are adapted from the Bureau's current operational forecasting models as described in section 2.3. The same climate predictors are applied at all lead times to

ensure a consistent model set up.

The full mathematical formulation of the BJP modelling approach is presented in Wang et al. (2009) and Wang and Robertson (2011). Here, we note some key features of the BJP modelling approach. The BJP models are able to effectively handle missing and non-concurrent records. The BJP models are based upon the multivariate normal distribution after allowing for data transformation using either the log-sinh (Wang et al., 2012) or Yeo-Johnson (Yeo and Johnson, 2000) transformations. If a set of variables follow a multivariate normal distribution then a subset of those variables also follows a multivariate normal distribution. Thus the many instances of missing data in streamflow records, as described in section 2.2, are easily handled. Several of the forecast locations experience ephemeral and intermittent flows, which can result in a probability mass for zero flows. This problem is handled in the BJP modelling approach by treating zero flows as censored data, meaning that the observations of zero flow are treated as being of unknown precise value equal to or below zero. Uncertainty in the model parameters due to the short data records is handled by inferring parameters using Markov Chain Monte Carlo methods (MCMC). Probabilistic (ensemble) forecasts are produced using conditional multivariate normal distribution equations. When predictor-predictand relationships are weak, the BJP modelling approach is designed to produce reliable forecasts that approximate climatology (i.e. frequency distribution of historically observed streamflow).

Some aspects of the BJP implementation used in this study vary compared with those published in Wang et al. (2009) and Wang and Robertson (2011). We use fixed transformation parameters together with simplified parameter reparameterisations for more straightforward numerical inference (Zhao et al., 2016) . The changes reflect our experience in achieving more robust and efficient BJP modelling.

## 3.2 Verification

In this study, we assess the performance of forecasts for the period from JFM 1982 to DJF 2011-12. A separate BJP model is established for each season, forecast location, and lead time. Across different years, the model parameter inference and forecast process is cross-validated using leave-five-years-out cross-validation. For each historical forecast event to be tested, the data points for the year to be forecast plus the subsequent four years are left out. The leave-five-years-out procedure is designed to account for strong persistence in streamflows, potentially over many years.

Forecasts from the BJP modelling approach are probabilistic. The continuous ranked probability score (CRPS; Matheson and Winkler, 1976) is used to assess full forecast probability distributions, therefore involving forecast ensemble spread as well as forecast accuracy. The CRPS for a given forecast and observation is defined as,

$$\text{CRPS} = \int \left[ F(y) - H(y - y_{obs}) \right]^2 dy \qquad (1)$$

where $y$ is the forecast variable, $y_{obs}$ is the observed value, $F(.)$ is the forecast cumulative distribution function (CDF) $H(.)$ is the Heaviside step function, which equals 0 if $y < y_{obs}$ and equals 1 otherwise. Model forecasts are compared to reference forecasts by calculating skill scores:

$$\text{CRPS skill score} = \frac{\overline{\text{CRPS}_{\text{ref}}} - \overline{\text{CRPS}}}{\overline{\text{CRPS}_{\text{ref}}}} \times 100 \qquad (\%) \tag{2}$$

where the overbar indicates averaging across a set of events. A climatology forecast is used as the reference in this study, although one may choose to use other reference forecasts. The CRPS skill score is positively oriented (whereas CRPS is negatively oriented). As a percentage, a maximum score of 100 is indicative of perfect forecasts. A score of 0 indicates no overall improvement compared to the reference forecast. A negative score indicates poor quality forecasts in the sense that a naïve climatology forecast is more skilful. CRPS skill scores are calculated for each catchment and season.

Reliability refers to the statistical agreement of forecast probabilities with observed relative frequencies of events, which can be checked using probability integral transforms (PITs). At each lead time, the PIT values are calculated from the BJP-generated ensemble forecasts for every forecast event, forecast location and season, and pooled in the analysis. The PIT represents the non-exceedance probability of observed streamflow obtained from the CDF of ensemble forecast. If the ensemble spread is appropriate and the forecasts are free of bias then observations will be contained within the forecast ensemble spread. Reliable forecasts are evidenced by PIT values that follow a uniform distribution between 0 and 1.

Sharpness refers to the concentration of the ensemble members and is a property of the forecast only. Forecast sharpness is desirable provided the forecast is reliable. A common measure of sharpness is the width or, more pertinently, the relative width of a forecast quantile range. Comparisons between forecasts can be made by comparing average widths or using a sharpness boxplot (Gneiting et al., 2007). In our study, we compare the quantile range widths of forecasts with lead time of 1–21 days with the quantile range widths of forecasts with 0-days lead time. Averages are taken for each catchment and season.

## 4 Results

### 4.1 Reliability

We first assess the reliability of the forecasts for the different lead times. Histograms of PIT values are constructed for each lead time from 0–14 days (Figure 3). For each lead time, the PIT histograms follow a similar distribution: approximately uniform but not perfectly so. There are a couple of reasons why the PIT histograms are not perfectly uniform. First, it is very difficult to model the shape of the forecast distributions perfectly, particularly under cross-validation. Second, a prior distribution over the data transformation parameters is influential in preventing very skewed distributions when there is limited data or many instances of zero flows. Thus, the model attempts to reliably reflect uncertainty, but cannot tightly fit the shape of the data in all cases.

On-the-whole, the PIT values suggest that the forecasts capture the range of observations sufficiently well and there is no strong evidence of bias. Therefore, the spread of the ensembles is deemed to be appropriate. However, note that the shape of the PIT histograms suggests that an alternative distribution, for example a fat-tailed distribution, may be able to fit the tails of

high-flow events better. The PIT histograms exhibit a similar shape for all lead times, meaning that the forecasts are similarly reliable across all lead times.

## 4.2 Accuracy

We now assess the accuracy skill of the forecasts for the different lead times. To assess forecast accuracy we calculate CRPS for each forecast location, season and lead time. It is possible to decompose the CRPS into components reflecting accuracy and reliability. Results in the previous section demonstrate that the forecasts are similarly reliable across all times. Therefore differences in the CRPS will mainly reflect differences in forecast accuracy. CRPS scores are in the original units of the data and therefore it is difficult to compare across forecast locations and seasons. CRPS skill scores, as defined in Equation (2),

however, measure the percentage reduction in error relative to a reference forecast, and therefore it is easier to compare relative skill levels across forecast locations, seasons and lead times. Hereafter, we consider only CRPS skill scores with modelled leave-five-years-out climatological distributions as reference forecasts unless otherwise specified.

CRPS skill scores for forecasts with 0-days lead time are plotted in Figure 4. Forecast locations form rows and target seasons form columns. The CRPS skill scores vary highly between forecast locations and seasons, due to variations in catchment

memory and climate predictability. The sources of predictability are not the main concern here though. Here, the aim is to analyse the changes in forecasting skill with increasing lead time. As an example, forecasts with 7-days lead time are plotted in Figure 5. By visual comparison, skill is overall reduced, however there are individual cases where skill increases. Instances of positive CRPS skill score differences are conceivable given the small sample size (30 years) and associated uncertainty in the skill score.  The patterns of skill across forecast locations and seasons are largely consistent at 0- and 7-days lead time.

Overall, increasing the forecast lead time to 7 days does not appear to significantly decrease forecasting skill. To investigate further, CRPS skill scores for 7-day lead time forecasts are calculated with the 0-day lead time forecasts as the reference rather than climatology (Figure 6).  The median CRPS skill score (considering all catchments and seasons) is -1.8. Catchment by catchment, the median CRPS skill score across all seasons ranges from -5.1 to 0.8, indicating that CRPS scores typically increase by less than about 5%, although larger increases in errors are observed in individual catchments and seasons.

Figure 7 illustrates for lead times of 0-, 7- and 14-days, the proportion of cases (catchments and seasons) where CRPS skill score thresholds are exceeded. CRPS skill scores between -5 and 5 are considered unskilful, below -5 is considered negative skill, and above 5 is considered skilful. The proportion of cases where certain CRPS skill score thresholds are exceeded decreases as the lead time increases. A CRPS skill score of -5 is exceeded in approximately 95%, 94% and 92% of cases for 0-, 7- and 14-days lead time respectively, indicating only a small proportion of cases exhibit skill worse than climatology. The

small difference in exceedance probabilities with increasing lead time is consistent with the knowledge that the BJP modelling approach should produce forecasts approximating climatology in the absence of any real forecasting skill.  Under stringent leave-five-years-out cross-validation, instances of skill negative skill can occur for various reasons related to aridity, poor catchment memory, extreme events and data problems. A CRPS skill score of 5 is exceeded in approximately 66%, 59% and

57% of cases for 0-, 7- and 14-days lead time respectively. The larger difference in exceedance probabilities suggests that the number of days lead time is important for skilful forecast locations and catchments.

To evaluate the change in forecast skill for each 1 day increase in lead time, we calculate the difference between CRPS skill scores for 1–14 and 21-day lead times with CRPS skill scores at lead time 0. For each lead time, the skill score differences for all forecast locations and seasons are pooled. Simplified boxplots of the CRPS skill score differences for each lead time are plotted (Figure 8). In each plot, the boxes represent the [0.25,0.75] and [0.10, 0.90] quantile ranges with the median marked by the line crossing the box. Outliers are ignored. The median of the boxplots indicates a clear monotonic trend, demonstrating that, on average, skill decreases as every 1 day forecast lead time increases. However, for individual forecast locations and seasons, the pattern of changes in skill is more variable. In some cases, the CRPS skill score can decrease by up to 18 percentage points (pp) as lead time increases up to 14 days. On the contrary, CRPS skill scores can increase by up to 10 pp as lead time increases up to 14 days. The trend of decreasing CRPS skill scores continues to 21-days lead time.

From Figure 4 and 5, it is evident that many forecast locations and seasons have low skill. We now partition the catchment and seasons into two groups based on CRPS skill scores at 0-days lead time. If the CRPS skill score for a catchment and season at 0-days lead time is > 5, the case is assigned to the skilful group. Otherwise, the case is assigned to the unskilful group. Figure 9 plots the change in forecast skill for each 1 day increase in lead time, considering only skilful cases. The reduction in skill is more marked for skilful cases compared with all cases. Figure 10 plots the change in forecast skill for each 1 day increase in lead time, considering only unskilful cases. The average reduction in CRPS skill scores is near zero, with variation mainly within ±5 pp. The results in Figure 9 (considering only skilful cases) are arguably the most important for analysing reductions in skill because the Bureau issues climatology forecasts when hindcast skill is poor. From Figure 9, at 7-days lead time the mean reduction in CRPS skill scores is about 4 pp, whereas at 14-days lead time it is about 6 pp. At 21-days the reduction in CRPS skill scores averages 7.5 pp with few cases associated with skill improvements.

## 4.3 Sharpness

Forecast sharpness is a property of the forecasts only and relates to the narrowness of the forecast ensemble spread. That is, a forecast with a narrower spread is sharper than a forecast with a wider spread. We assess how forecast sharpness changes as forecast lead time is increased. For each forecast event, the [0.1,0.9] quantile range of 1–21 day lead time forecasts is divided by the [0.1,0.9] quantile range for the 0-day lead time forecast. An average is then calculated for each catchment and season, which is termed the average relative width (ARW). Simplified boxplots of the ARW for each lead time are plotted (Figure 11). In each plot, the boxes represent the [0.25,0.75] and [0.05, 0.95] quantile ranges of ARW with the median marked by the line crossing the box. Sharpness is seen to decrease gradually as forecast lead time is increased. At 7-days lead time, the forecasts are typically only marginally wider than at 0-days lead time, with the median ARW approximately 1.02. However, the forecasts can be considerably wider in some cases, with the values of ARW up to 1.2 quite possible in some catchments and seasons. At 21-days lead time the median ARW increases to 1.08.

**5 Discussion**

The results demonstrate that forecasts are similarly reliable for all lead times from 0-14 days. Hence, from the perspective of reliability, any forecast lead time is similarly optimal. Forecasts released with 7- or 14-days lead time will be similarly reliable to forecasts released with 0 days lead time. Therefore, considerations for optimal forecast lead time do not need to be based on reliability.

The results demonstrate that mean skill, averaged across all forecast locations and seasons, decreases monotonically for lead times from 0-14 days and the trend continues to 21-days lead time. However, the reduction in skill ought to be distinguished for skilful catchments, since the skill for difficult to model catchments with poor skill remains largely unchanged and the Bureau's forecasts are forced to a climatological distribution when skill is very low or negative. From the perspective of accuracy skill, the monotonicity of decreasing skill scores with lead time suggests that the optimum forecast lead time is the shortest lead time that allows for other factors including the number of days required for forecast preparation, dissemination, comprehension and application.

As discussed in the introduction, the Bureau currently issues forecasts at least 7 days into the forecast season. Thus 7 days can be regarded as the current time needed for forecast preparation and dissemination.  Much of the delay in preparing the current operational forecasts is attributable to delays in obtaining various climate data sets and quality streamflow data. The approach taken in this study of relying solely on daily SST data for climate predictors significantly reduces the burden in preparing climate indices. Preliminary daily OISST data sets are available within 1 day, although they are subject to revision for up to 14 days. Thus the necessary climate indices can theoretically be ready for inclusion in forecasting models within 1-2 days. As discussed in the introduction, quality-controlled streamflow data can take up to 3 days to enter the Bureau's forecasting database. It is therefore expected that the minimum predictor data preparation time is 3 days.

Forecast and communication strategy production is a process that takes 1-2 days. Thus for the Bureau to consider releasing forecasts prior to the beginning of the target season, it would be a safe choice to prepare forecasts with 7-day lead time. Compared with forecasts with 0-days lead time, the mean reduction in CRPS skill scores in skilful cases is approximately 4 pp (Figure 9), which is likely to be tolerated by forecast users in exchange for earlier forecast release. However, it is to be reasonably expected that CRPS skill scores will reduce by up to 10–15 pp in some instances (Figure 9). The significant reduction in skill scores at 21-days lead time (Figure 9) highlights the importance of short-lead time forecasts for Australian catchments and confirms that simply switching to a 1-month ahead forecast system is undesirable.

A better, more flexible operational forecasting system for Australia could be built upon a flexible strategy that allows for any number of lead times (in days). Such a system allows for multiple forecast runs prior to forecast release. To reduce forecast preparation time, while striving for optimal skill, a communication strategy for the forecasts can be developed based on a preliminary forecast run with, for example, 7-days lead time. The final forecast release could subsequently be based on a shorter lead time, e.g. 4-days. Furthermore, it sometimes happens that unexpected heavy rains fall within the period of forecast

generation and forecast release. If significant events occur that change the hydrological outlooks dramatically, having the option to reissue forecasts could benefit users.

The discussion thus far has considered only the forecast preparation and dissemination time. As identified previously, optimal forecast lead time for operational decision making depends on other factors including the time needed for comprehension and application. That is, forecast users need time to understand the forecasts and the likely impact on their operations. The process of understanding may include further sophisticated modelling using streamflow forecasts as inputs. For water managers, lead times of a few days may be sufficient to assimilate the forecast information. For other operators, longer lead-time forecasts may be preferred. In fact, the optimal forecast lead time for operational decision making across a range of industries is likely to vary. It therefore remains a research questions whether water forecasting services need to evolve to cater for the needs of different (sophisticated) water forecast users.

## 6 Conclusion

Currently the Bureau of Meteorology releases seasonal streamflow forecasts approximately 7 days into the forecast target season. In this study we develop seasonal streamflow forecasting models with 0–14 and 21 days lead time to demonstrate that it is possible to release skilful operational forecasts ahead of the commencement of the forecast period. Forecasts were produced for 23 of the Bureau of Meteorology's seasonal streamflow forecast locations, using the Bayesian joint probability modelling approach. The forecasting models were constructed similarly to the Bureau's official models, using climate predictors and initial catchment condition predictors. Climate predictors were adapted to be based on SST predictors so that daily SST data sets could be adopted.

The skill and reliability of the 0-14 day lead time forecasts were assessed. Reliability was found to be similar for all forecast lead times. Average skill reduces monotonically for each 1 day that forecast lead time increases. For forecasts with 7-days lead time, the mean reduction in CRPS skill scores is small, approximately 4 percentage points, although skill score differences within a range of +5 to -15 are possible. For forecasts with 14-days lead time, the mean reduction in CRPS skill scores is approximately 6 percentage points. In correspondence with decreasing forecast skill, forecast sharpness reduces slightly as forecast lead time is increased. The reductions in skill are very likely to be tolerated by forecast users in exchange for forecasts released ahead of the commencement of the forecast target season. Particularly as the Bureau moves towards a monthly forecasting service, timelier forecast release is going to become critical.

## Acknowledgements

This research was funded by the Water Information Research and Development Alliance (WIRADA), a partnership between CSIRO and the Bureau of Meteorology.

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

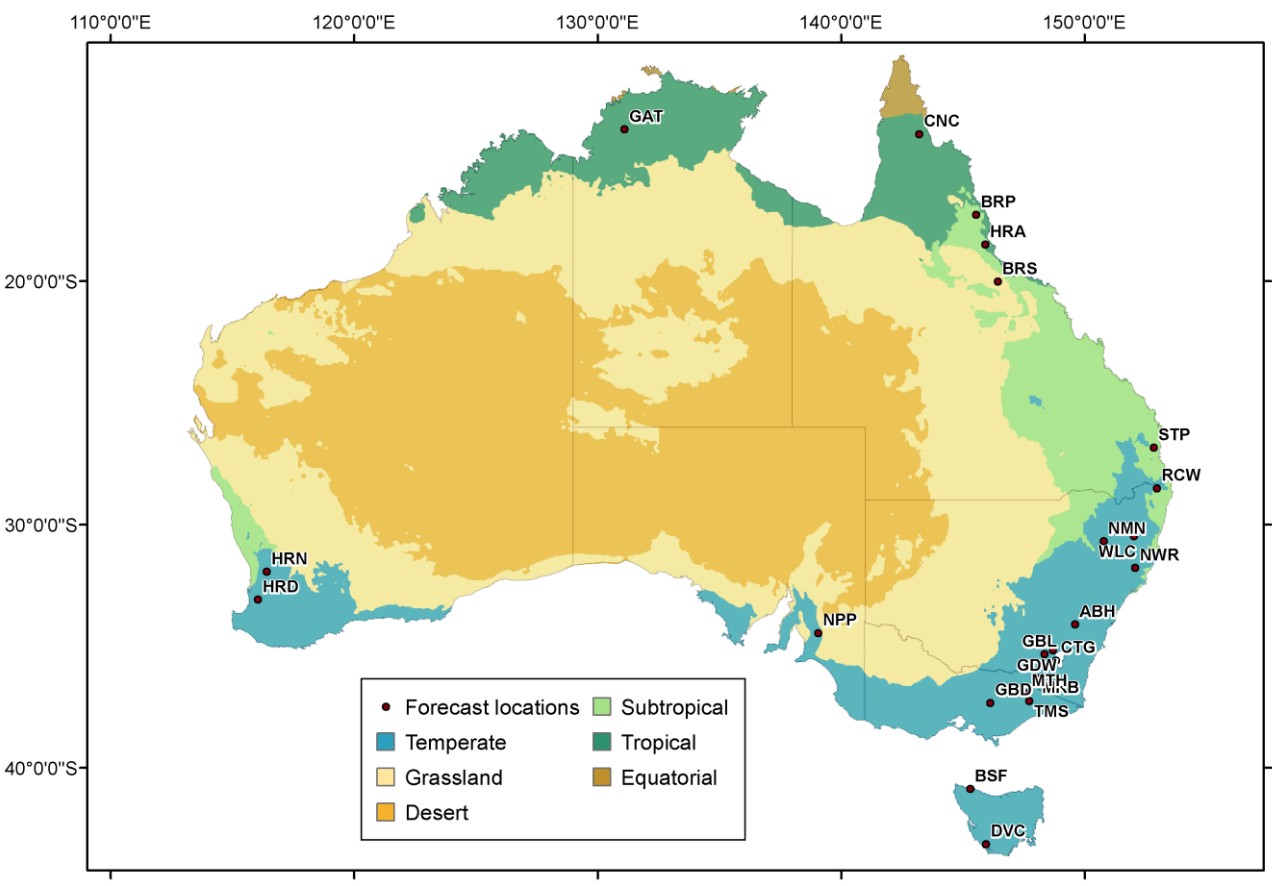

**Figure 1: The 23 forecast locations and their climate zones. The locations are current Bureau of Meteorology seasonal streamflow forecast locations.**

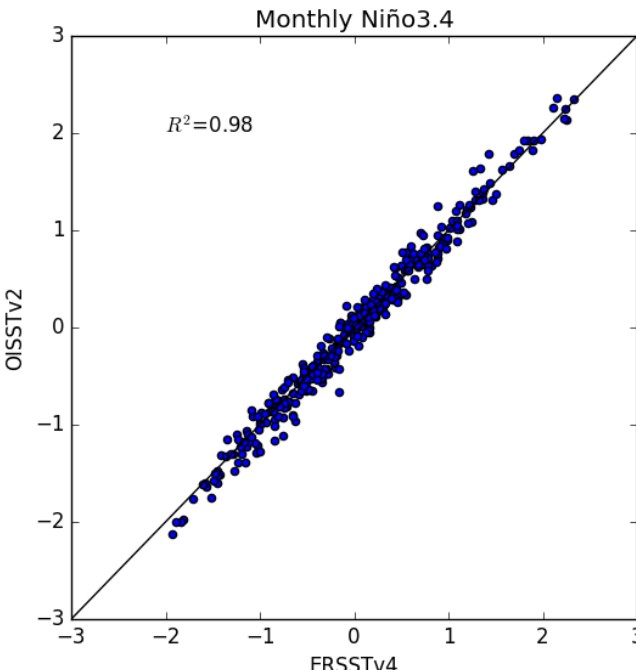

**Figure 2: Scatterplot demonstrating the strong relationship between monthly Niño3.4 anomalies calculated from monthly ERSST v4 data and daily OISST v2 data.**

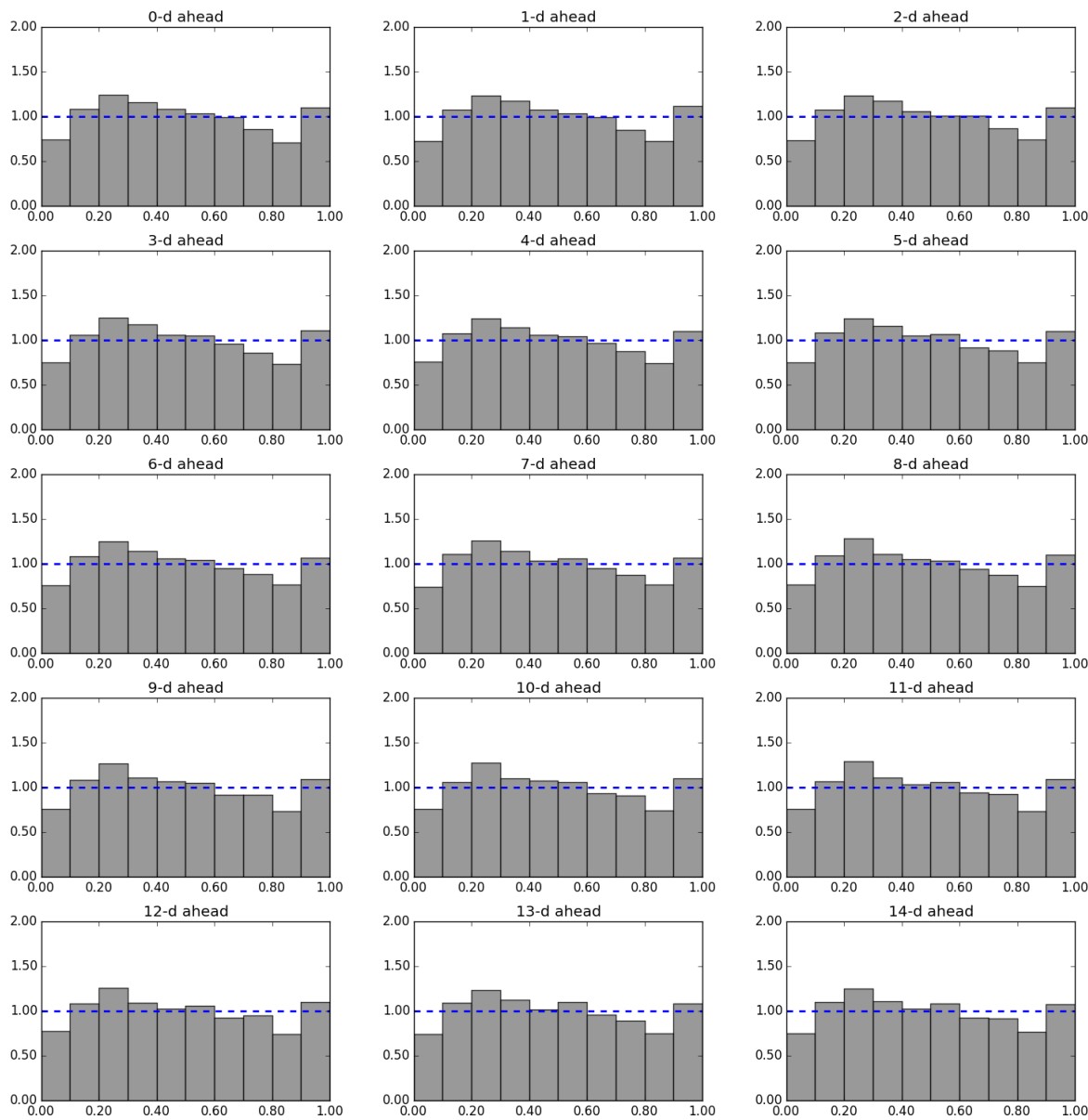

**Figure 3: Figure 3: PIT histograms of seasonal streamflow forecasts for lead times ranging from 0–14 days (the x-axis of the plot represents PIT value and the y-axis normalised frequency). The blue dotted line marks the expected frequency.**

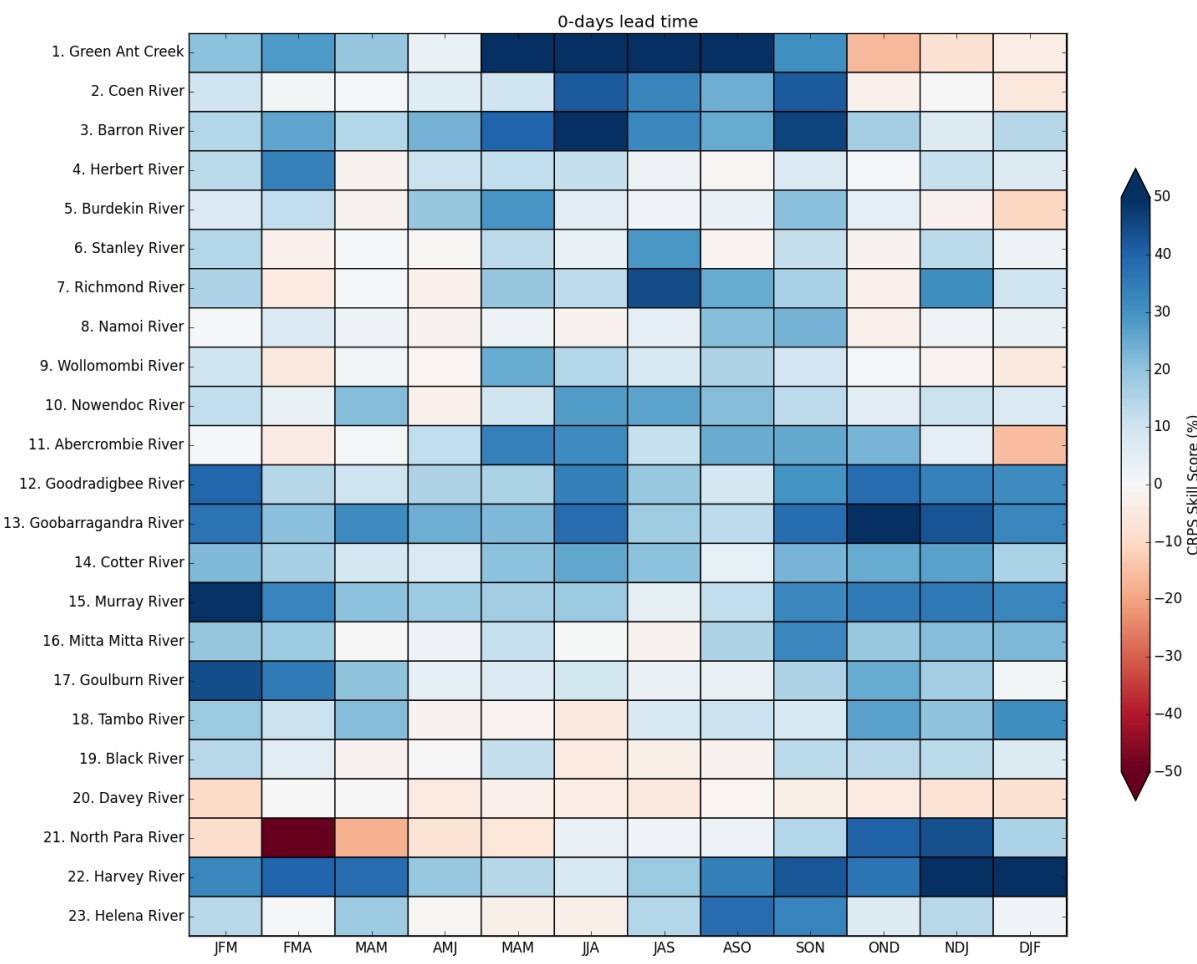

**Figure 4: CRPS skill scores for each catchment and target season at 0-days lead time. Skill scores are relative to climatology. Leave-five-years-out cross validation is applied for the period 1982–2011.**

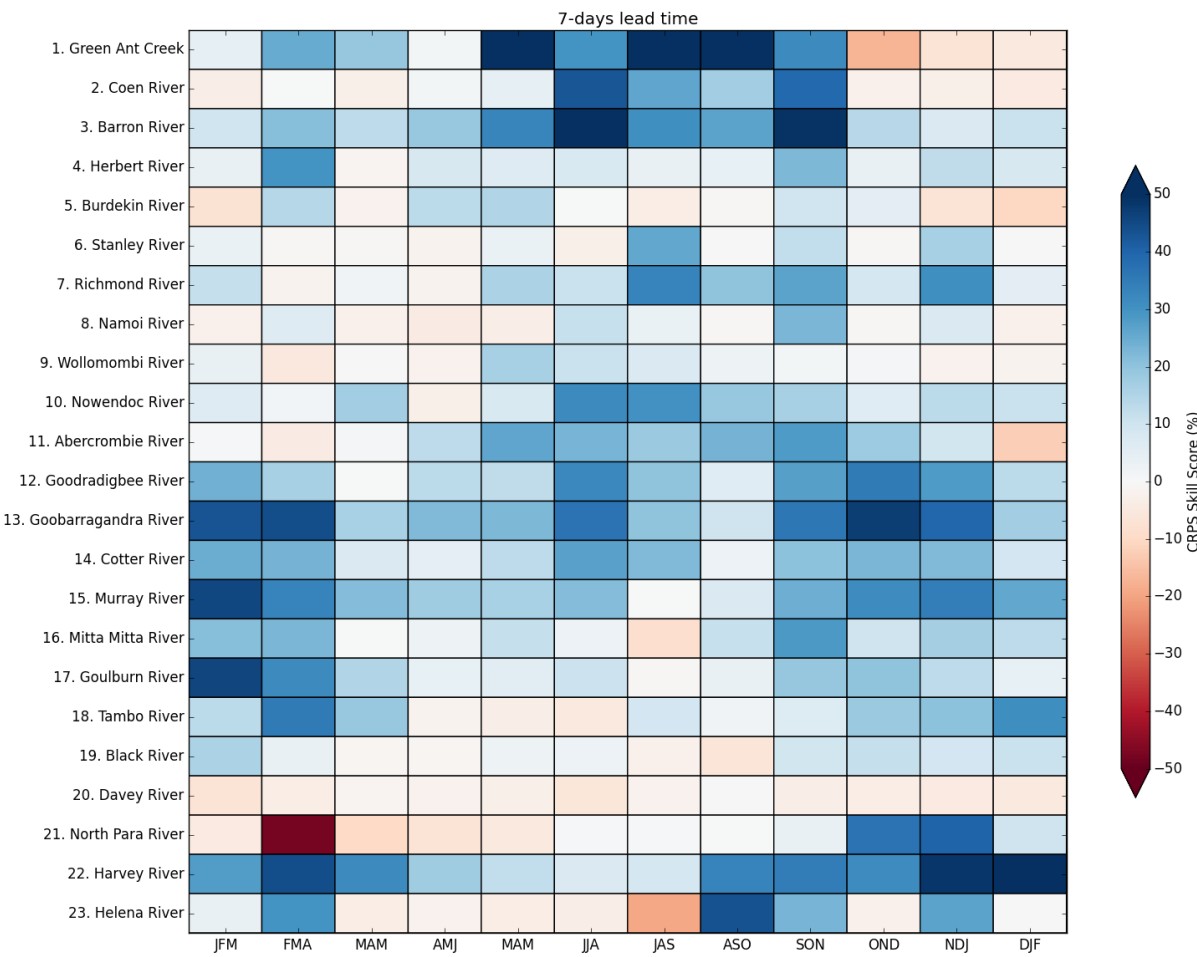

**Figure 5: CRPS skill scores for each catchment and target season at 7-days lead time. Skill scores are relative to climatology. Leave-five-years-out cross validation is applied for the period 1982–2011.**

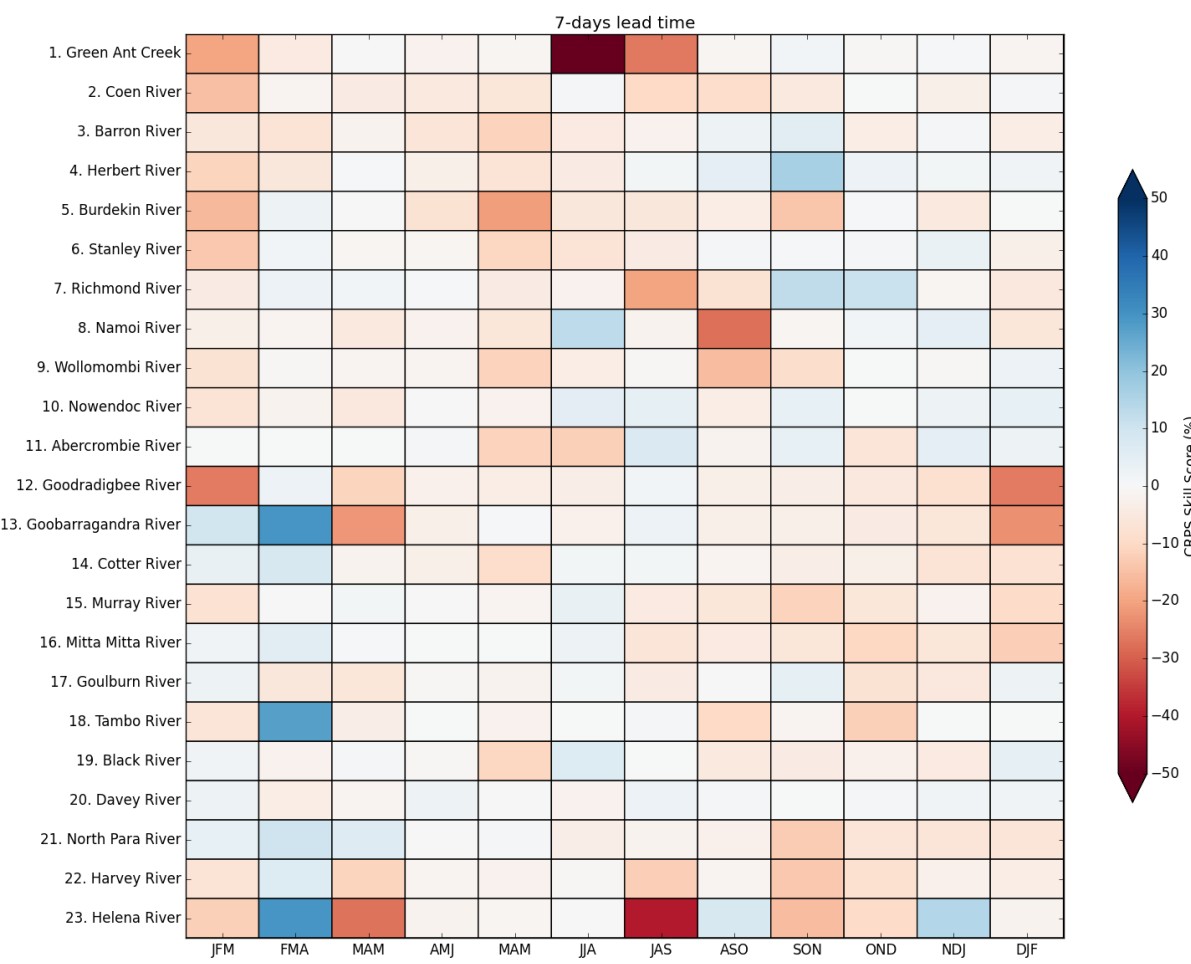

**Figure 6: CRPS skill scores for each catchment and target season at 7-days lead time. Skill scores are relative to 0-day lead time forecasts.**

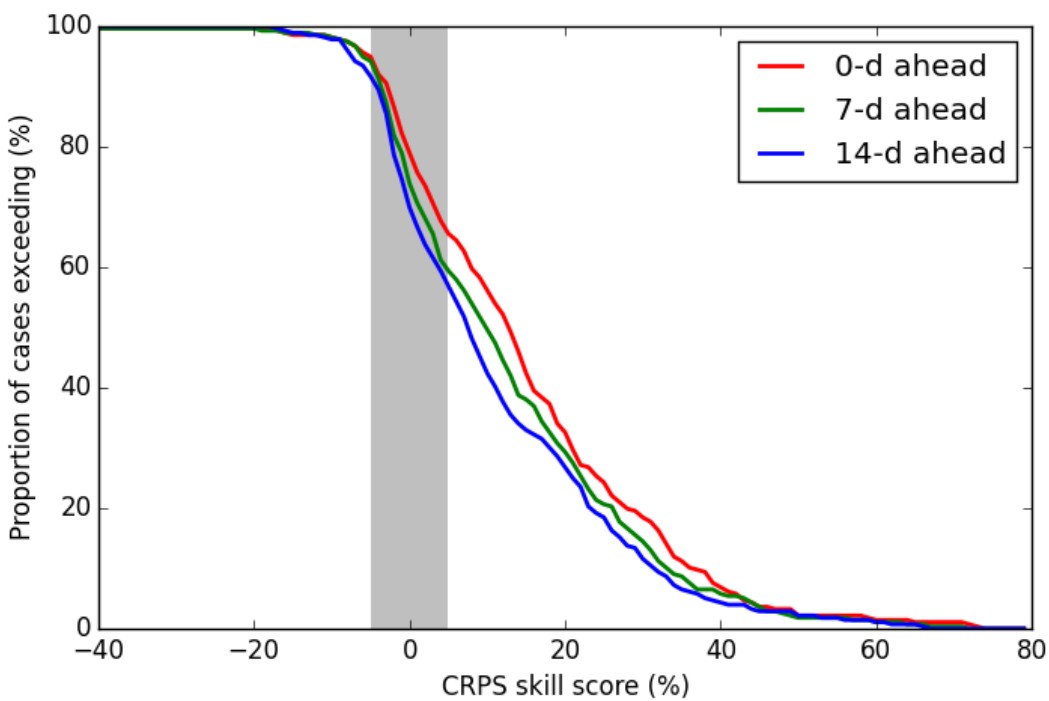

**Figure 7: Proportion of cases exceeding CRPS skill scores thresholds for lead times of 0-, 7- and 14-day lead times. For each lead time, the CRPS skill scores for each forecast location and season have been pooled. The grey shaded region indicates neutral skill (±5).**

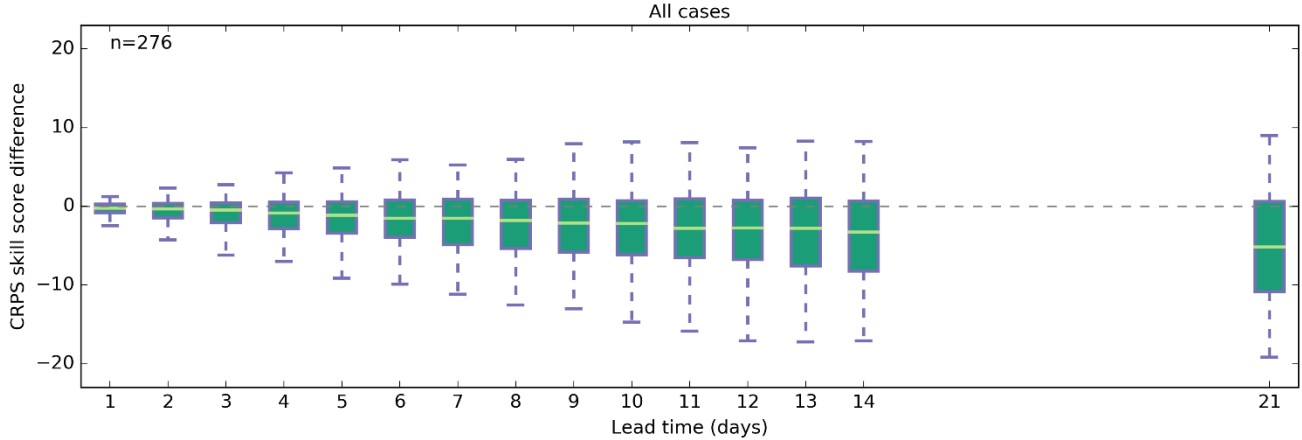

**Figure 8: Box-plots of CRPS skill score differences between forecasts at 1–21 days lead time and lead time 0. The boxes capture the median and the [0.25,0.75] and [0.05,0.95] quantile ranges. For each lead time, the CRPS skill score differences for every forecast location and season have been pooled. n is the number of cases per boxplot (23 locations x 12 seasons = 276 cases).**

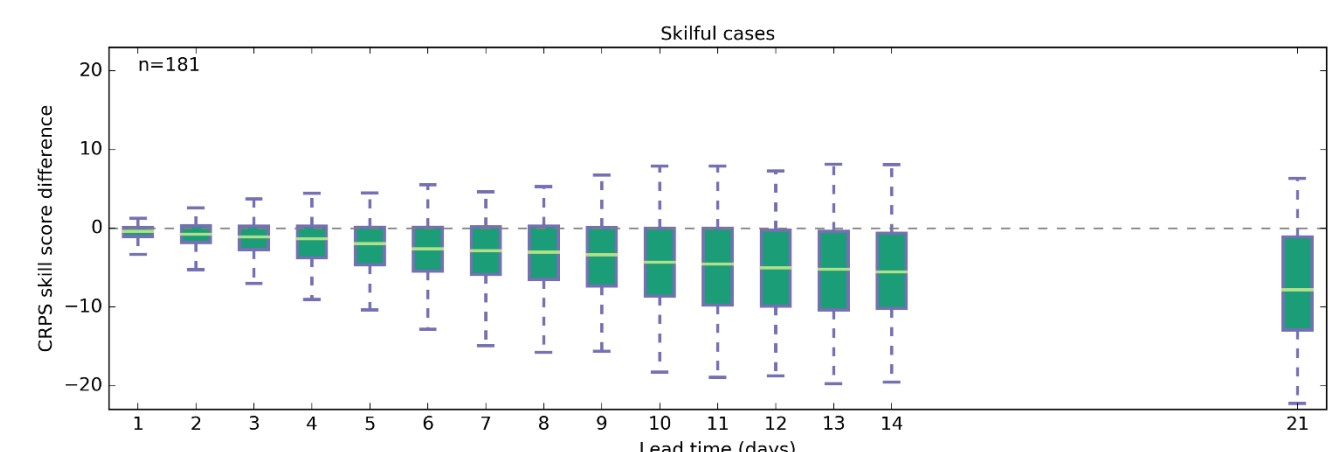

**Figure 9: As for Figure 8, except considering only cases where the 0-day lead time forecast is skilful (CRPS skill score > 5)**

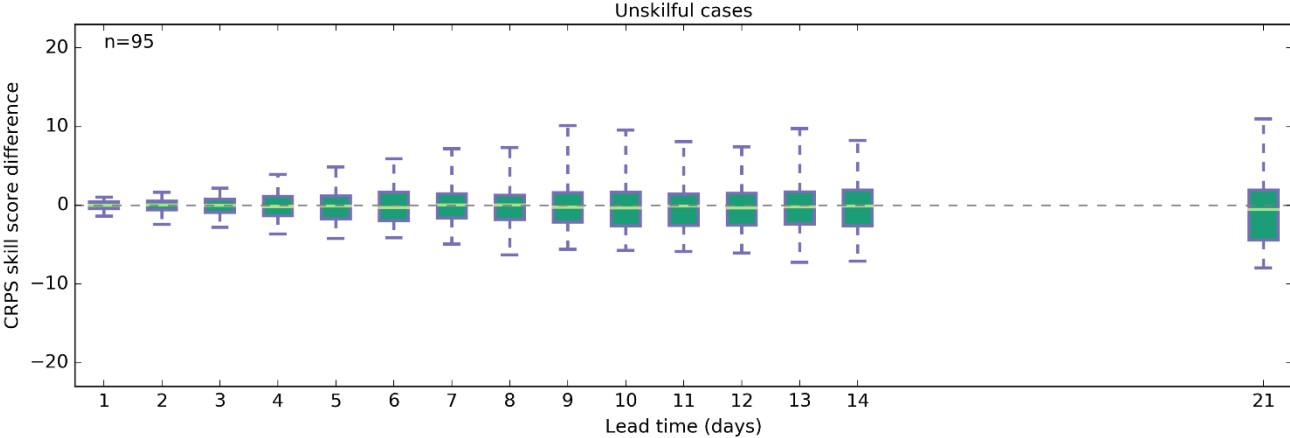

**Figure 10: As for Figure 8, except considering only cases where the 0-day lead time forecast is not skilful (CRPS skill score <= 5)**

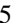

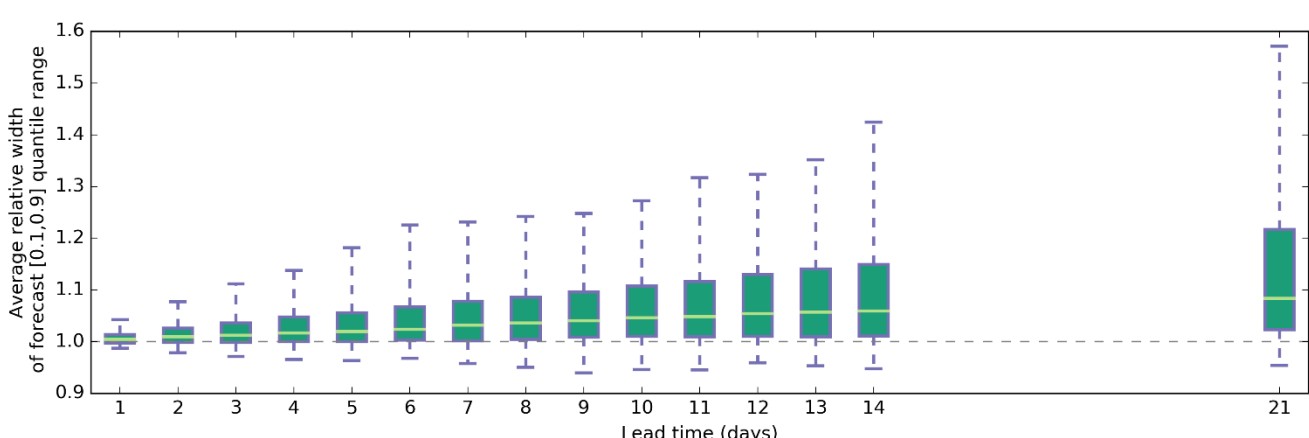

**Figure 11: Box-plots of the average relative width of the [0.1, 0.9] forecast quantile range of forecasts at 1–21 days lead time. The relativity is to forecasts at lead time 0. The average relative width is calculated for each forecast location and season. Cases where a forecast quantile range is 0 are omitted.**

Table 1: Catchment information for the 23 forecast locations

| Catchment No. | Short ID | Long name | State | Area (km2) | Lon | Lat |
|---|---|---|---|---|---|---|
| G8140161 | GAT | Green Ant Creek at Tipperary | NT | 416 | 131.1 | -13.7 |
| 922101B | CNC | Coen River above Coen Racecourse | QLD | 170 | 143.2 | -13.9 |
| 110003A | BRP | Barron River above Picnic Crossing | QLD | 239 | 145.5 | -17.3 |
| 116006B | HRA | Herbert River above Abergowrie | QLD | 7486 | 145.9 | -18.5 |
| 120002 | BRS | Burdekin River above Sellheim | QLD | 36230 | 146.4 | -20 |
| 143303A | STP | Stanley River above Peachester | QLD | 102 | 152.8 | -26.8 |
| 203005 | RCW | Richmond River above Wiangaree | QLD | 712 | 153 | -28.5 |
| 419005 | NMN | Namoi River above North Cuerindi | QLD | 2532 | 150.8 | -30.7 |
| 206014 | WLC | Wollomombi River above Coninside | NSW | 377 | 152 | -30.5 |
| 208005 | NWR | Nowendoc River above Rocks Crossing | NSW | 1893 | 152.1 | -31.8 |
| 412066 | ABH | Abercrombie River above Hadley No.2 | NSW | 1631 | 149.6 | -34.1 |
| 410024 | GDW | Goodradigbee River above Wee Jasper | NSW | 990 | 148.7 | -35.2 |
| 410057 | GBL | Goobarragandra River above Lacmalac | NSW | 668 | 148.4 | -35.3 |
| 410730 | CTG | Cotter River above Gingera | NSW | 130 | 148.8 | -35.6 |
| 401012 | MRB | Murray River above Biggara | NSW | 1257 | 148.1 | -36.3 |
| 401203 | MTH | Mitta Mitta River above Hinnomunjie | VIC | 1518 | 147.6 | -37 |
| 405219 | GBD | Goulburn River above Dohertys | VIC | 700 | 146.1 | -37.3 |
| 223202 | TMS | Tambo River above Swifts Creek | VIC | 899 | 147.7 | -37.3 |
| 14213 | BSF | Black River at South Forest | TAS | 319.1 | 145.3 | -40.9 |
| 473 | DVC | Davey River above D/S Crossing Rv | TAS | 698 | 146 | -43.1 |
| A5050517 | NPP | North Para River at Penrice | SA | 121 | 139.1 | -34.5 |
| 613002 | HRD | Harvey River above Dingo Road | WA | 148 | 116 | -33.1 |
| 616013 | HRN | Helena River at Ngangaguringuring | WA | 316 | 116.4 | -31.9 |

