# Peer review of "Optimising Seasonal Streamflow Forecast Lead Time for Operational Decision Making in Australia"

_Hydrology and Earth System Sciences, 2016_

## Referee Comment (RC1) · Anonymous Referee #1 · 10 Jun 2016

General Comments This paper proposes a new forecast technique for water supply forecasting in Australia designed to allow for more frequent updates as well as to improve forecast accuracy and to make the forecasts more timely. Currently operational forecasts are released 7 days into a forecast target period. This proposed technique would include the entire forecast period in the future. Overall, the paper was logically organized and easy to follow. The verification metrics used were applicable to the problem and applied in a logical manner. The results supported the conclusions in the paper well. I am curious why the authors did not investigate incorporation of weather prediction into their algorithm. This would seem to be a major area ripe for improvement as the forecast skill at the weather timescale (0-2 weeks into the future) is much larger than at climate time scales and is ever increasing as computer power continues to grow. It was striking that there was no discussion on how forecast utilization. I recommend including information on how the current forecasts are utilized and how the new forecasts might improve application. Specific comments [page 3; lines 9-11] How is the undesirability of forecasts beyond 1 one month consistent with the premise of this paper of improving 3 month forecasts? [page 3; line 13] What is N equal to? [page 4; line 23] is daily updated subsurface temperatures really necessary for this method? Does it change that much from one day to the next?

---

## Referee Comment (RC2) · B. Klein (Referee) · 14 Jun 2016

General Comments:

The paper covers the important topic of forecast release time of monthly to seasonal forecasts. Many available observation products used for statistical forecasting, such as mean monthly SST of the last month, are generally available several days after the beginning of the forecast period. Additional time is needed for data control, the generation of the seasonal forecasts and the development of key messages and other communication products.

In the Australian forecast system presented here the forecasts are generally issued with a lag of 7 days. New data products based on daily SST from NOAA are tested for a timelier release of the forecast. The verification of the results is straightforward

and the methods used for verification, PIT and CRPSS, are suitable for this topic. The results show only a small degradation of average skill for forecasts with 7-days lead time compared to the current method. This is a good message as timelier release of forecast increases its potential value in decision making. Another advantage of the presented method is that the predictands are not limited to calendar months any more. In theory when using daily data a forecast of the next 28 days could be released every week.

The paper is of great importance, well written and easy to understand. It fits nicely the topic of the special issue and should be foreseen for publication. The only thing I am struggling with a little bit is that the current version of the paper is between a technical study and a scientific paper. Additional references of methods used for statistical forecasting and the differences compared to the BJP and the advantages of the BJP should be presented in the introduction. An additional interesting verification metric would be the sharpness of the predictive uncertainty of the different lead times compared to the original method (not only in the combined measure CRPSS). As the training data set of the available daily SST data product (starting 1982) is smaller than the training data set of the monthly SST data (starting 1854 but also depending on the length of the streamflow observation record) it is expected that the parameter and the total uncertainty of the BJP predictions are larger for the predictors with a smaller observation period.

Specific Comments:

p 4, l 5: Explain shortly the main characteristics of the different runoff regimes for non-Australian river experts. Probably add the regions of the different runoff regimes to Figure 1.

p 4, l 6: Add range of catchment areas considered: "... ranging from 102 to 36 230 km$^2$..."

p 4, l 10: Length of streamflow observation records? This is important to get an impression of the number of data points used for parameter estimation of the BJP (see

general comment).

p 4, l 15: Make clear that the predictands are still the three-month totals starting at the beginning of each month.

p 4, l 25 – l 30: Is there a relationship between subsurface ocean temperatures and SOI lagged by two / three months with the predictant? These predictors could still be used in the system with lead times up to 28 days.

p 8, l 8: Add % to the CRPSS values. In many other applications a maximum CRPSS of 1 (100% in your case) is used. This could be a little bit confusing.

---

## Referee Comment (RC3) · Anonymous Referee #3 · 14 Jun 2016

Summary

This paper proposes to increase the lead time at which operational forecasts of three-month streamflow volumes can be issued by the Bureau of Meteorology in Australia. Currently, the data collection and processing cause the forecasts to be sometimes issued in the second week of the forecast period. The development of sub-seasonal forecasts in the region prompts the need for timelier forecasts. In this paper, the authors first present a forecasting method that would allow forecasts to be issued several days prior to the forecast period. Then, they investigate the relationship between forecast lead time and forecast quality.

The authors show that being able to issue forecasts several days prior to the forecast period can be achieved without any loss in forecast reliability. Nevertheless, accuracy

decreases as the lead time increases. The optimal trade-off between forecast accuracy and forecast lead time must then be decided on. Forecasts with lead times shorter than 7 days may offer an acceptable loss in accuracy as compared to the gain in anticipation time.

General comments

The topic is an interesting one and allows to look at forecast quality within an operational context, with timing constraints The paper is very well written and is easy to follow. The proposed figures are also easy to read and well illustrate the paper. A catchment set is used to obtain relatively general results and the forecast evaluation criteria are relevant. The detailed comments are only minor suggestions and questions I had when reading the manuscript.

Detailed comments

Throughout the paper, "N-days lead time" and "N-month lead time" were confusing at first, probably due to the "N". Could you maybe reformulate or add a small sentence in the first occurrences to define the terms?

In Section 2.3, have you compared the performance of the current forecasting system with that of the proposed system? Do you have an idea of their difference in skill, if the forecasts are both run at lead time 0?

Page 8, Lines 8-9: Is there a specific reason why you chose -5 and 5 as thresholds?

Page 8, Lines 12-13: Could you maybe add a short sentence to further explain this? For instance, is the number of cases below -5 sufficient to assert this?

Page 9, Lines 26-27: This sentence is, I think, very important in this paper, as it links forecast quality with operational expectations and requirements. I am just curious: have the authors further investigated the topic? How would you discuss / How have you discussed the acceptable loss in quality with operational managers? Is there a way to weigh the loss in skill more pragmatically, and relate it to the gain in anticipation

time?

Figure 4 and 5: It is interesting to put in parallel these figures with Figure 1 and Table 1. But how are the catchments ordered? I could not find it in the text, nor when looking at Table 1.

Figure 5: If I understand correctly, the objective of this figure is to show that the patterns between 0-day lead time and 7-days lead time are quite similar. Have you thought of showing the skill score of the 7-days lead time computed with the 0-day lead time as reference to highlight the differences between Figures 4 and 5? I understand that Figures 6 and 7 go in this direction but Page 8 Lines 3-6 could be illustrated in that way as well.

[Figure]

---

## Author Comment (AC1) · 25 Jul 2016

**Responses to Reviewer 1**

General Comments

This paper proposes a new forecast technique for water supply forecasting in Australia designed to allow for more frequent updates as well as to improve forecast accuracy and to make the forecasts more timely. Currently operational forecasts are released 7 days into a forecast target period. This proposed technique would include the entire forecast period in the future. Overall, the paper was logically organized and easy to follow. The verification metrics used were applicable to the problem and applied in a logical manner. The results supported the conclusions in the paper well.

I am curious why the authors did not investigate incorporation of weather prediction into their algorithm. This would seem to be a major area ripe for improvement as the forecast skill at the weather timescale (0-2 weeks into the future) is much larger than at climate time scales and is ever increasing as computer power continues to grow.

Response: The reviewer makes a valid point about the potential for weather forecasts to improve the skill of streamflow forecasts up to two weeks into the future. It is an active area of research within our research group however it is not within the scope of this study. This study considers the forecasting of three-month seasonal streamflow totals using statistical methods where most of the skill comes from initial catchment conditions and limited predictability stems from climate.

It was striking that there was no discussion on how forecast utilization. I recommend including information on how the current forecasts are utilized and how the new forecasts might improve application.

Response: We agree. We are able to include more specific examples about how streamflow forecasts are used by water managers in Australia. We have prepared several paragraphs on 1. Overall forecast use 2. More specific uses, and 3. Advantage of earlier forecasts. These paragraphs will be included in the revised manuscript.

Specific comments

[page 3; lines 9-11] How is the undesirability of forecasts beyond 1 one month consistent with the premise of this paper of improving 3 month forecasts?

Response: We believe our comment is not inconsistent because it is specifically about the time between forecast issue and the first day of the forecast period. We can make this point clearer.

[page 3; line 13] What is N equal to?

Response: N is used to indicate any number of days or months. We will make this clearer by changing the symbol or being more explicit.

[page 4; line 23] is daily updated subsurface temperatures really necessary for this method? Does it change that much from one day to the next?

Response: The subsurface ocean temperatures will not change much from day to day and, strictly speaking, it would not be essential to have daily updates to use subsurface temperatures in our approach. We believe our comments of subsurface and atmospheric predictors are distracting and can be safely removed.

---

## Author Comment (AC2) · 25 Jul 2016

**Responses to Reviewer 2**

General Comments

The paper covers the important topic of forecast release time of monthly to seasonal forecasts. Many available observation products used for statistical forecasting, such as mean monthly SST of the last month, are generally available several days after the beginning of the forecast period. Additional time is needed for data control, the generation of the seasonal forecasts and the development of key messages and other communication products. In the Australian forecast system presented here the forecasts are generally issued with a lag of 7 days. New data products based on daily SST from NOAA are tested for a timelier release of the forecast. The verification of the results is straightforward and the methods used for verification, PIT and CRPSS, are suitable for this topic. The results show only a small degradation of average skill for forecasts with 7-days lead time compared to the current method. This is a good message as timelier release of forecast increases its potential value in decision making. Another advantage of the presented method is that the predictands are not limited to calendar months any more. In theory when using daily data a forecast of the next 28 days could be released every week. The paper is of great importance, well written and easy to understand. It fits nicely the topic of the special issue and should be foreseen for publication.

The only thing I am struggling with a little bit is that the current version of the paper is between a technical study and a scientific paper. Additional references of methods used for statistical forecasting and the differences compared to the BJP and the advantages of the BJP should be presented in the introduction.

Response: The reviewer makes a useful point and we agree it would be beneficial to include more information about BJP and how it fits in the frame of statistical forecasting methods. We are very willing to enhance the introduction to include more information and references in this regard.

An additional interesting verification metric would be the sharpness of the predictive uncertainty of the different lead times compared to the original method (not only in the combined measure CRPSS). As the training data set of the available daily SST data product (starting 1982) is smaller than the training data set of the monthly SST data (starting 1854 but also depending on the length of the stream- flow observation record) it is expected that the parameter and the total uncertainty of the BJP predictions are larger for the predictors with a smaller observation period.

Response: Sharpness is a useful measure and we agree it could be useful to have a distinct analysis of sharpness in our paper. We are prepared to investigate how forecast sharpness changes as forecast lead time is increased. We would measure sharpness as width of forecast intervals relative to reference forecasts (e.g. climatology or lead 0). If the results are conclusive and informative, we will be happy to include them in the paper.

The point about forecast sharpness and length of record is very valid. In our study, we set up models as consistently as possible, using a fixed period, to enable comparisons between forecast locations and seasons. In practice, the weather Bureau may use longer periods of streamflow data to establish the models at individual forecast locations and thus achieve improved forecast sharpness. This point will be added to the discussion.

Specific Comments:

p 4, l 5: Explain shortly the main characteristics of the different runoff regimes for non-Australian river experts. Probably add the regions of the different runoff regimes to Figure 1.

Response: Figure 1 will be updated to include climate zones to which runoff regimes are closely related.

p 4, l 6: Add range of catchment areas considered: ". . . ranging from 102 to 36 230 km2 . . ."

Response: Easily done

p 4, l 10: Length of streamflow observation records? This is important to get an impression of the number of data points used for parameter estimation of the BJP (see general comment).

Response: Thanks for pointing out the omission. The data period coincides with the verification period. We will add this information to section 2.2 on streamflow data.

p 4, l 15: Make clear that the predictands are still the three-month totals starting at the beginning of each month.

Response: Easily done

p 4, l 25 – l 30: Is there a relationship between subsurface ocean temperatures and SOI lagged by two / three months with the predictant? These predictors could still be used in the system with lead times up to 28 days.

Response: Yes, however we cannot guarantee nor quantify the usefulness of this approach. We believe our comments of subsurface and atmospheric predictors are distracting and can be safely removed.

p 8, l 8: Add % to the CRPSS values. In many other applications a maximum CRPSS of 1 (100% in your case) is used. This could be a little bit confusing.

Response: Easily done

---

## Author Comment (AC3) · 25 Jul 2016

**Responses to Reviewer 3**

Summary

This paper proposes to increase the lead time at which operational forecasts of three month streamflow volumes can be issued by the Bureau of Meteorology in Australia. Currently, the data collection and processing cause the forecasts to be sometimes issued in the second week of the forecast period. The development of sub-seasonal forecasts in the region prompts the need for timelier forecasts. In this paper, the authors first present a forecasting method that would allow forecasts to be issued several days prior to the forecast period. Then, they investigate the relationship between forecast lead time and forecast quality. The authors show that being able to issue forecasts several days prior to the forecast period can be achieved without any loss in forecast reliability. Nevertheless, accuracy decreases as the lead time increases. The optimal trade-off between forecast accuracy and forecast lead time must then be decided on. Forecasts with lead times shorter than 7 days may offer an acceptable loss in accuracy as compared to the gain in anticipation time.

General comments

The topic is an interesting one and allows to look at forecast quality within an operational context, with timing constraints The paper is very well written and is easy to follow. The proposed figures are also easy to read and well illustrate the paper. A catchment set is used to obtain relatively general results and the forecast evaluation criteria are relevant. The detailed comments are only minor suggestions and questions I had when reading the manuscript.

Detailed comments

Throughout the paper, "N-days lead time" and "N-month lead time" were confusing at first, probably due to the "N". Could you maybe reformulate or add a small sentence in the first occurrences to define the terms?

Response: N is used to indicate any number of days or months. We can certainly make this clearer by using a different symbol or being more explicit.

In Section 2.3, have you compared the performance of the current forecasting system with that of the proposed system? Do you have an idea of their difference in skill, if the forecasts are both run at lead time 0?

Response: A very interesting question and one we pondered how to address when designing our study. Ultimately, we decided there are too many confounding factors to include a direct comparison in our paper.

However, in Australia, most of the forecast skill is acquired through the initial catchment condition predictors. Indeed, many models have no climate predictors. Therefore we expect the skill of the two systems to be relatively on par at 0 lead time.

Page 8, Lines 8-9: Is there a specific reason why you chose -5 and 5 as thresholds?

Response: These thresholds are essentially rules-of-thumb based on years of experience and some bootstrap experiments in previous studies. We can clarify this point.

Page 8, Lines 12-13: Could you maybe add a short sentence to further explain this? For instance, is the number of cases below -5 sufficient to assert this?

Response: The BJP modelling approach should theoretically produce a reliable climatology-like forecast in the absence of predictor-predictand relationships. In practice, under our stringent leave-five-years-out cross-validation, instances of skill negative skill can occur for various reasons related to short data records, low flows, a high density of zero flows, limited catchment memory etc.

We can certainly add a sentence or two to explain reasons for the small proportion of cases with negative skill. Additionally, we are able to include more information in the methodology section about BJP's propensity to produce climatology-like forecasts in the absence of a forecasting relationship.

Page 9, Lines 26-27: This sentence is, I think, very important in this paper, as it links forecast quality with operational expectations and requirements. I am just curious: have the authors further investigated the topic? How would you discuss / How have you discussed the acceptable loss in quality with operational managers? Is there a way to weigh the loss in skill more pragmatically, and relate it to the gain in anticipation time?

Response: The study was motivated by the demands of water managers to have timelier forecasts. The weather Bureau will continue to engage with operational managers to determine the best outcome for them. On pages 9-10, the benefit of having a flexible system is discussed, with regard to meeting the needs of operational managers.

Figure 4 and 5: It is interesting to put in parallel these figures with Figure 1 and Table 1. But how are the catchments ordered? I could not find it in the text, nor when looking at Table 1.

Response: The order of catchments in Table 1 and Figures 4 and 5 is not consistent – thanks for detecting. In the revision, we intend to order the catchments clockwise from northeast Australia to southwest Western Australia.

Figure 5: If I understand correctly, the objective of this figure is to show that the patterns between 0-day lead time and 7-days lead time are quite similar. Have you thought of showing the skill score of the 7-days lead time computed with the 0-day lead time as reference to highlight the differences between Figures 4 and 5? I understand that Figures 6 and 7 go in this direction but Page 8 Lines 3-6 could be illustrated in that way as well.

Response: It is certainly feasible to calculate the skill scores as suggested, to show the relative difference in error between the 0-day and 7-day lead time forecasts. However, as rightly pointed out, the differences in skill are illustrated by subsequent figures. Our preference is to leave the figures unchanged as they are effectively conveying the messages written into the text.

---

## Author Response (AR1)

**Responses to Reviewer 1**

General Comments

This paper proposes a new forecast technique for water supply forecasting in Australia designed to allow for more frequent updates as well as to improve forecast accuracy and to make the forecasts more timely. Currently operational forecasts are released 7 days into a forecast target period. This proposed technique would include the entire forecast period in the future. Overall, the paper was logically organized and easy to follow. The verification metrics used were applicable to the problem and applied in a logical manner. The results supported the conclusions in the paper well.

I am curious why the authors did not investigate incorporation of weather prediction into their algorithm. This would seem to be a major area ripe for improvement as the forecast skill at the weather timescale (0-2 weeks into the future) is much larger than at climate time scales and is ever increasing as computer power continues to grow.

Response: The reviewer makes a valid point about the potential for weather forecasts to improve the skill of streamflow forecasts up to two weeks into the future. It is an active area of research within our research group however it is not within the scope of this study. This study considers the forecasting of three-month seasonal streamflow totals using statistical methods where most of the skill comes from initial catchment conditions and limited predictability stems from climate.

It was striking that there was no discussion on how forecast utilization. I recommend including information on how the current forecasts are utilized and how the new forecasts might improve application.

Response: We agree. We are able to include more specific examples about how streamflow forecasts are used by water managers in Australia. We have prepared several paragraphs on 1. Overall forecast use 2. More specific uses, and 3. Advantage of earlier forecasts. These paragraphs are included near the beginning of the introduction (P1 L29 – P2 L13).

Specific comments

[page 3; lines 9-11] How is the undesirability of forecasts beyond 1 one month consistent with the premise of this paper of improving 3 month forecasts?

Response: We explain that the primary source of skill is initial catchment conditions and catchments can have limited memory or short response times (P3 L24-25). It is reasonable to expect that a three-month-total forecast issued today will have much higher skill than a three-month-total forecast issued four weeks ago.

[page 3; line 13] What is N equal to?

Response: "N-days" no longer appears in the manuscript. We have replaced all instances with more precise descriptions.

[page 4; line 23] is daily updated subsurface temperatures really necessary for this method? Does it change that much from one day to the next?

Response: The subsurface ocean temperatures will not change much from day to day and, strictly speaking, it would not be essential to have daily updates to use subsurface temperatures in our approach. We believe the comment about not having daily subsurface temperatures was distracting and so it has been removed.

**Responses to Reviewer 2**

General Comments

The paper covers the important topic of forecast release time of monthly to seasonal forecasts. Many available observation products used for statistical forecasting, such as mean monthly SST of the last month, are generally available several days after the beginning of the forecast period. Additional time is needed for data control, the generation of the seasonal forecasts and the development of key messages and other communication products. In the Australian forecast system presented here the forecasts are generally issued with a lag of 7 days. New data products based on daily SST from NOAA are tested for a timelier release of the forecast. The verification of the results is straightforward and the methods used for verification, PIT and CRPSS, are suitable for this topic. The results show only a small degradation of average skill for forecasts with 7-days lead time compared to the current method. This is a good message as timelier release of forecast increases its potential value in decision making. Another advantage of the presented method is that the predictands are not limited to calendar months any more. In theory when using daily data a forecast of the next 28 days could be released every week. The paper is of great importance, well written and easy to understand. It fits nicely the topic of the special issue and should be foreseen for publication.

The only thing I am struggling with a little bit is that the current version of the paper is between a technical study and a scientific paper. Additional references of methods used for statistical forecasting and the differences compared to the BJP and the advantages of the BJP should be presented in the introduction.

Response: The reviewer makes a useful point and we agree it would be beneficial to include more information about BJP and how it fits in the frame of statistical seasonal streamflow forecasting methods in Australia. We have added a number of references to the introduction and indicated why BJP was selected for operational use (P3 L32 – P4 L2).

An additional interesting verification metric would be the sharpness of the predictive uncertainty of the different lead times compared to the original method (not only in the combined measure CRPSS). As the training data set of the available daily SST data product (starting 1982) is smaller than the training data set of the monthly SST data (starting 1854 but also depending on the length of the stream- flow observation record) it is expected that the parameter and the total uncertainty of the BJP predictions are larger for the predictors with a smaller observation period.

Response:  Sharpness is a useful measure and we agree it could be useful to have a distinct analysis of sharpness in our paper. We have investigated how forecast sharpness changes as forecast lead time is increased. We measure sharpness as the width of forecast intervals relative to lead time 0 forecasts. The results are now included and discussed in the paper, and the methods updated accordingly (Figure 11 and P7 L13-17 and P9 L23–32)

The point about forecast sharpness and length of record is very valid. In our study, we set up models as consistently as possible, using a fixed period, to enable comparisons between forecast locations and seasons. In practice, the weather Bureau may use longer periods of streamflow data to establish the models at individual forecast locations and thus achieve improved forecast sharpness. Whilst an important point from an operational point of view, we haven't discussed this matter in the paper.

Specific Comments:

p 4, l 5: Explain shortly the main characteristics of the different runoff regimes for non-Australian river experts. Probably add the regions of the different runoff regimes to Figure 1.

Response: Figure 1 has been updated to include climate zones to which runoff regimes are closely related. The text has been updated to reflect the change (P4 L14-15).

p 4, l 6: Add range of catchment areas considered: ". . . ranging from 102 to 36 230 km2 . . ."

Response: Easily done (P4 L16).

p 4, l 10: Length of streamflow observation records? This is important to get an impression of the number of data points used for parameter estimation of the BJP (see general comment).

Response: Thanks for pointing out the omission. The data period coincides with the verification period. We have added this information to section 2.2 on streamflow data (P4 L18).

p 4, l 15: Make clear that the predictands are still the three-month totals starting at the beginning of each month.

Response: Easily done (P4 L24-25).

p 4, l 25 – l 30: Is there a relationship between subsurface ocean temperatures and SOI lagged by two / three months with the predictant? These predictors could still be used in the system with lead times up to 28 days.

Response: Yes, however we cannot guarantee nor quantify the usefulness of this approach. We believe the comment about not having frequently updated SOI was distracting and so it has been removed.

p 8, l 8: Add % to the CRPSS values. In many other applications a maximum CRPSS of 1 (100% in your case) is used. This could be a little bit confusing.

Response: Thanks for pointing out the omission. We have corrected the methods section to point out that the CRPS skill scores should be interpreted as percentages (P7)

**Responses to Reviewer 3**

Summary

This paper proposes to increase the lead time at which operational forecasts of three month streamflow volumes can be issued by the Bureau of Meteorology in Australia. Currently, the data collection and processing cause the forecasts to be sometimes issued in the second week of the forecast period. The development of sub-seasonal forecasts in the region prompts the need for timelier forecasts. In this paper, the authors first present a forecasting method that would allow forecasts to be issued several days prior to the forecast period. Then, they investigate the relationship between forecast lead time and forecast quality. The authors show that being able to issue forecasts several days prior to the forecast period can be achieved without any loss in forecast reliability. Nevertheless, accuracy decreases as the lead time increases. The optimal trade-off between forecast accuracy and forecast lead time must then be decided on. Forecasts with lead times shorter than 7 days may offer an acceptable loss in accuracy as compared to the gain in anticipation time.

General comments

The topic is an interesting one and allows to look at forecast quality within an operational context, with timing constraints The paper is very well written and is easy to follow. The proposed figures are also easy to read and well illustrate the paper. A catchment set is used to obtain relatively general results and the forecast evaluation criteria are relevant. The detailed comments are only minor suggestions and questions I had when reading the manuscript.

Detailed comments

Throughout the paper, "N-days lead time" and "N-month lead time" were confusing at first, probably due to the "N". Could you maybe reformulate or add a small sentence in the first occurrences to define the terms?

Response: "N-days" no longer appears in the manuscript. We have replaced all instances with more precise descriptions.

In Section 2.3, have you compared the performance of the current forecasting system with that of the proposed system? Do you have an idea of their difference in skill, if the forecasts are both run at lead time 0?

Response: A very interesting question and one we pondered how to address when designing our study. Ultimately, we decided there are too many confounding factors to include a direct comparison in our paper.

However, in Australia, most of the forecast skill is acquired through the initial catchment condition predictors. Indeed, many models have no climate predictors. Therefore we expect the skill of the two systems to be relatively on par at 0 lead time.

Page 8, Lines 8-9: Is there a specific reason why you chose -5 and 5 as thresholds?

Response: These thresholds are essentially rules-of-thumb based on years of experience and some bootstrap experiments in previous studies.

Page 8, Lines 12-13: Could you maybe add a short sentence to further explain this? For instance, is the number of cases below -5 sufficient to assert this?

Response: The BJP modelling approach should theoretically produce a reliable climatology-like forecast in the absence of predictor-predictand relationships. In practice, under our stringent leave-five-years-out cross-validation, instances of skill negative skill can occur for various reasons related to short data records, low flows, a high density of zero flows, limited catchment memory etc. We have updated the text accordingly (P8 L31–33). We have also included more information in the methodology section about BJP's propensity to produce climatology-like forecasts in the absence of a forecasting relationship (P6 L11–13).

Page 9, Lines 26-27: This sentence is, I think, very important in this paper, as it links forecast quality with operational expectations and requirements. I am just curious: have the authors further investigated the topic? How would you discuss / How have you discussed the acceptable loss in quality with operational managers? Is there a way to weigh the loss in skill more pragmatically, and relate it to the gain in anticipation time?

Response: The study was motivated by the demands of water managers to have timelier forecasts and so the weather Bureau will continue to engage with operational managers to determine the best outcome for them. We have revised the introduction to include more information about the usage of the forecasts in various agencies and the potential benefits of timelier forecasts (P1 L29 – P2 L13).

Figure 4 and 5: It is interesting to put in parallel these figures with Figure 1 and Table 1. But how are the catchments ordered? I could not find it in the text, nor when looking at Table 1.

Response: The order of catchments in Table 1 and Figures 4 and 5 was not consistent – thanks for detecting. We now order the catchments clockwise from northeast Australia to southwest Western Australia.

Figure 5: If I understand correctly, the objective of this figure is to show that the patterns between 0-day lead time and 7-days lead time are quite similar. Have you thought of showing the skill score of the 7-days lead time computed with the 0-day lead time as reference to highlight the differences between Figures 4 and 5? I understand that Figures 6 and 7 go in this direction but Page 8 Lines 3-6 could be illustrated in that way as well.

Response: It is certainly feasible to calculate the skill scores as suggested, to show the relative difference in error between the 0-day and 7-day lead time forecasts. We have included the suggested figure and updated the text accordingly (Figure 6; P8 L20-24). However, as rightly pointed out, the differences in skill are illustrated by subsequent figures.

[revised manuscript text omitted]

$$\cancel{CRPS = \frac{1}{T}\sum_{t=1}^{T}\int\left[F(y^t) - H(y^t - d_y^t)\right]^2 dy^t} \quad \mathrm{CRPS} = \int\left[F(y) - H(y - y_{obs})\right]^2 dy$$

(1)

 where $\cancel{y^t}$ $y$ is the forecast variable  $\cancel{d_y^t}$ $y_{obs}$ is the  observed value, $F(.)$ is the forecast cumulative distribution function (CDF) $H(.)$ is the Heaviside step function, which equals 0 if $\cancel{y^t < d_y^t}$ $y < y_{obs}$ and equals 1 otherwise.  Model forecasts are compared to ref,  reference forecasts by calculating skill scores:

$$\cancel{CRPS_{SkillScore} = \frac{CRPS_{ref} - CRPS}{CRPS_{ref}}} \quad \mathrm{CRPS\ skill\ score} = \frac{\overline{\mathrm{CRPS_{ref}}} - \overline{\mathrm{CRPS}}}{\mathrm{CRPS_{ref}}} \times 100 \quad (\%)$$

(2)

[revised manuscript text omitted]